# METCAM Is a Potential Biomarker for Predicting the Malignant Propensity of and as a Therapeutic Target for Prostate Cancer

**DOI:** 10.3390/biomedicines11010205

**Published:** 2023-01-13

**Authors:** Jui-Chuang Wu, Guang-Jer Wu

**Affiliations:** 1Department of Chemical Engineering, Chung Yuan Christian University, Taoyuan City 32023, Taiwan; 2Research Center for Circular Economy, Chung Yuan Christian University, Taoyuan City 32023, Taiwan; 3Department of Bioscience Technology, Chung Yuan Christian University, Taoyuan City 32023, Taiwan; 4Department of Microbiology and Immunology, Emory University School of Medicine, Atlanta, GA 30322, USA

**Keywords:** METCAM, biomarker of prostate cancer, Western blot and ELISA, traditional and modified LFIA, magnetic beads-enriched array display, magnetic beads-purified serum, biotinylated antibodies, nano-gold conjugated antibodies, streptavidin, tumorigenesis and metastasis driver, therapeutic target for prostate cancer

## Abstract

Prostate cancer is the second leading cause of cancer-related death worldwide. This is because it is still unknown why indolent prostate cancer becomes an aggressive one, though many risk factors for this type of cancer have been suggested. Currently, many diagnostic markers have been suggested for predicting malignant prostatic carcinoma cancer; however, only a few, such as PSA (prostate-specific antigen), Prostate Health Index (PHI), and PCA3, have been approved by the FDA. However, each biomarker has its merits as well as shortcomings. The serum PSA test is incapable of differentiating prostate cancer from BPH and also has an about 25% false-positive prediction rate for the malignant status of cancer. The PHI test has the potential to replace the PSA test for the discrimination of BPH from prostate cancer and for the prediction of high-grade cancer avoiding unnecessary biopsies; however, the free form of PSA is unstable and expensive. PCA3 is not associated with locally advanced disease and is limited in terms of its prediction of aggressive cancer. Currently, several urine biomarkers have shown high potential in terms of being used to replace circulating biomarkers, which require a more invasive method of sample collection, such as via serum. Currently, the combined multiple tumor biomarkers may turn out to be a major trend in the diagnosis and assessment of the treatment effectiveness of prostate cancer. Thus, there is still a need to search for more novel biomarkers to develop a perfect cocktail, which consists of multiple biomarkers, in order to predict malignant prostate cancer and follow the efficacy of the treatment. We have discovered that METCAM, a cell adhesion molecule in the Ig-like superfamily, has great potential regarding its use as a biomarker for differentiating prostate cancer from BPH, predicting the malignant propensity of prostate cancer at the early premalignant stage, and differentiating indolent prostate cancers from aggressive cancers. Since METCAM has also been shown to be able to initiate the spread of prostate cancer cell lines to multiple organs, we suggest that it may be used as a therapeutic target for the clinical treatment of patients with malignant prostate cancer.

## 1. Current State of Clinical Diagnosis of Malignant Prostate Cancer

Prostate cancer is the most commonly diagnosed cancer and the second leading cause of cancer death in males world-wide [1]. In most patients, prostatic cancer is localized within the prostate gland and poses no threat to their lives. Commonly, most prostate cancer patients die with this cancer, not from this cancer; thus, there is no need for treatment [2]. However, in about 10% patients, prostatic cancer may develop into a malignant state, wherein patients often succumb to death within one year [2,3,4]. To date, regardless of many years of intense research, how indolent, localized tumors become aggressive cancers is not clear [3,4]. At most, some risk factors have been defined, but their mechanisms are not understood. Moreover, the achievement of a precise prognosis of the cancer is not easy in clinical practice. The usual criteria for clinical prostate cancer are the results from the combination tests, which include the initial serum PSA (total PSA) test, abnormal results from a digital rectal examination (DRE), and the follow-up tests of either the more invasive needle biopsy or the less invasive—yet more expensive—Multi-Parametric MRI (MP-MRI). Since the results of digital rectal examination are not reliable, this practice will be replaced by an imaging method in developed countries. To avoid over-diagnosis and overtreatment, most modern research has focused on the discovery of a better biomarker to replace the serum PSA (total PSA) test and the use of an imaging method to replace biopsies for an accurate diagnosis of cancer and for the detection of recurrence. To date, many diagnostic markers have been suggested for predicting malignant prostatic carcinoma cancer; however, only a few, such as PSA (prostate-specific antigen), Prostate Health Index (PHI), and PCA3, have been approved by FDA [5].

The serum PSA (total PSA) test has been used for many years and is still used in developing countries; however, its use results in over-diagnosis and over-treatment, because the test has a 20–25% false diagnosis rate. PSA, a serine protease, is a prostate-specific antigen, but not a prostate-cancer-specific antigen [4]; thus, the test (with a cut-off point of >4 ng/mL) cannot accurately predict the malignancy of this cancer and fails to distinguish indolent cancer from aggressive cancer. Furthermore, the test cannot differentiate prostate cancer from BPH or prostatitis (which is an inflammation caused by infection of pathogens). For the past few years, major efforts have been focused on improving the PSA test because of its prostate-specificity. One of these efforts concerns the determination of the velocity of serum PSA within a certain time period; however, the major limitation of this method is that it cannot differentiate prostate cancer from BPH and prostatitis. Another method is to determine the ratio of free/total PSA; accordingly, a low free/total PSA (f/tPSA) ratio has been used to guide diagnosis in patients with PSA values in the “grey” zone (between 4 and 10 ng/mL) [6]. However, a f/tPSA ratio lower than the cut-off value has only been able to predict less than 50% of malignancies, leading to the over-performance of biopsies. Further attempts have been made to improve the poor diagnostic performance of the f/tPSA ratio by studying and exploring other PSA molecular forms as biomarkers; in particular, [-2]proPSA, whose production is selectively increased in cancer and is significantly associated with high-grade cancer (Gleason score ≥ 7) as revealed upon radical prostatectomy. The final product of these endeavors is the PHI test. The PHI test is a multifactorial mathematical combination of PSA, fPSA, and [-2]proPSA; it produces a risk index for the positive biopsy of high-grade cancer. Accordingly, a higher [-2]proPSA/fPSA ratio associated with a high tPSA level may suggest a diagnosis of clinically significant prostate cancer. Eventually, PHI was approved by the FDA in 2012 for men over 50 years of age with negative DRE and PSA in the grey zone (between 4 and 10 ng/mL). Thus, PHI has emerged as the best test with which to accurately differentiate the fatal aggressive cancers from the non-fatal indolent ones. However, it has two drawbacks: the free PSA form, [-2]proPSA, is unstable, and the cost of the related test is somewhat expensive [6]. Regardless, the use of a single PSA test is gradually fading, and it will eventually be replaced with a test of the combined multiple tumor biomarkers, which is becoming a major trend in the diagnosis and assessment of the treatment effectiveness of prostate cancer [4].

Another circulatory biomarker that has been used for diagnosis is PCA3 [7]. However, PCA3 has shown a controversial correlation with aggressiveness. In international guidelines, PCA3 is only recommended for patients undergoing repeated biopsy. When PCA3 is used together with the PSA test, it may be used to predict the presence of a malignant cancer, but it is useless for an indolent cancer [6,7]. At initial biopsies, both PCA3 and PHI were accurate predictors of cancer with no significant difference between them, suggesting that both have a comparable ability to discriminate benign and malignant conditions. In comparison to PCA3, PHI prevents a higher number of unnecessary biopsies without sacrificing the identification of high-grade cancer. PHI outperforms PCA3 in terms of the ability to predict tumor volume ≥ 0.5 mL, Gleason score ≥ 7, and tumor stage [6,7]. The combination of PCA3 and PHI was able to significantly improve cancer identification compared to the use of only one biomarker [6,7]. Many other biomarkers have been proposed for diagnosis; however, there is still a long way to go before they are approved by FDA [8].

In addition to the circulating markers, several urinary biomarkers have been developed, for which the following markers are well-supported by evidence and more widely used: PCA3 [7], TMPRSS2–ERG fusion gene product [9], 8-OHdG (8-Hydroxy-2-Deoxyguanosine), and 8-iso-PGF2α (8-Iso-Prostaglandin F2α) [10]. Regarding the use of PCA3 and TMRPSS2–ERG as urine biomarkers for prostate cancer, the following conclusions have been reached: For both PCA3 and TMRPSS2–ERG, post-DRE urine tests were developed. Among the urinary tests, PCA3 is recommended for patients undergoing repeated biopsy and for the identification of the risk of a positive biopsy for high-grade cancer; this is probably due to its controversial correlation with aggressiveness [6,7]. In a prospective multicenter study performed on 516 patients, TMRPSS2–ERG was not significantly correlated with the aggressiveness but the absence of malignancy. TMPRSS2–ERG had an independent, additional predictive value compared to PCA3 and the ERSPC (The European Randomized Study of Screening for Prostate Cancer) risk calculator parameters with respect to the prediction of a biopsy’s Gleason score and clinical tumor stage. Furthermore, TMPRSS2–ERG had prognostic value, whereas PCA3 had not. Since the sensitivity of PCA3 increased when it was combined with TMPRSS2–ERG, implementing the novel urinary biomarker panel PCA3 and TMPRSS2–ERG into clinical practice would lead to a considerable reduction in the number of prostate biopsies [6,9].

Regarding the use of 8-OHdG and 8-iso-PGF2α as urine biomarkers for prostate cancer, the following points have been concluded: The stable urinary metabolite, 8-OHdG, is a key biomarker of in vivo oxidative DNA damage. The urinary 8-iso-PGF2α could serve as a noninvasive biomarker for a reliable index of cyclooxygenase-catalyzed inflammation [10]. Elevated levels of both biomarkers are associated with the onset and progression of prostate cancer [10]. A liquid chromatography–tandem mass spectrometry method was used to reveal the enhanced values of these two major indices of oxidative stress damage in the urine samples of prostate cancer patients. The urinary levels of these two biomarkers were measured before and after RARP (Robot-Assisted Radical Prostatectomy), which is superior to open radical prostatectomy in terms of oncological outcomes, has a lower risk of positive surgical margin and a higher likelihood of the preservation of continence in high-risk settings, and normalizes the indices of oxidative stress. Both urinary markers were found to be higher than the control groups (sex- and age-matched healthy subjects) before RARP but lower than the control groups after RARP, indicating that the successful surgical excision of the prostate corrected these elevated oxidative indices in the prostate cancer patients. The above results support the notion that 8-OHdG and 8-Iso-PGF2α measurements in urine can help to predict radicality (and possibly local recurrence as well) following prostate cancer surgery, despite the fact that the biomarkers lack tissue specificity. In the near future, large-sized sample studies and long-term follow ups are needed to further validate these urinary biomarkers for use in the early prevention and successful clinical treatment of prostate cancer [10].

Besides the use of biomarkers, a needle biopsy is performed to provide the Gleason score (namely, the pathological stage) before a radical prostatectomy or a radiotherapy is performed. However, a biopsy may not capture all cases with clinical prostate cancer because most prostatic cancers are multi-focal, while the pathological grades found in needle biopsies often fail to represent the entire malignant potential of the tumor [2,3]. Thus, the costly Multi-Parametric MRI (MP-MRI) method may eventually be used to replace invasive biopsies in the future, especially in developed countries [6].

Since the use of combined multiple tumor biomarkers appears to be a major trend in the diagnosis and assessment of the treatment effectiveness of prostate cancer [4], there is still a need to search for more novel biomarkers so as to develop a perfect cocktail consisting of multiple biomarkers in order to predict malignant prostate cancer and to follow the efficacy of clinical treatment. Accordingly, METCAM may be a biomarker to meet this need, as illustrated below.

## 2. Human METCAM (huMETCAM) Is Overly Expressed in Prostate Cancer Tissues and in TRAMP, Suggesting That It May Serve as a Biomarker for Predicting the Malignant Potential of Prostatic Tumors

HuMETCAM is an integral membrane protein. It is also a cell adhesion molecule (CAM) belonging to the immunoglobulin-like gene superfamily. From its protein structure (Figure 1), we suggest that it is capable of executing many typical functions of CAMs: cell-to-cell interactions, cell-to-extracellular interactions, communicating with many intracellular signaling pathways, and regulating the social behaviors of the cells [11,12].

Initially, METCAM was shown to be abundantly expressed in most malignant melanoma tissues; consequently, it was named MUC18, because the authors thought it might be a mucin-like protein since it was expressed on the cellular membrane [11]. The protein was originally assumed to be only expressed in melanoma; thus, it was also named MelCAM [13] or MCAM [14]. Later, MUC18 was also shown to be expressed in endothelial cells and was named S-endo-1 and CD146 [15]. For a trivial reason, the protein also bears another name: A32 [16]. All the above alternative names did not reflect its protein structure and biological functions; thus, in 2005, we renamed it METCAM [17], a metastasis-regulating CAM. This is because, from our own research, we found that it plays a dual role in the malignant progression of many epithelial tumors [18]. The protein is expressed in about eight normal cells and many other cancers [19]. We also found that it is expressed in normal tissues, such as ovarian epithelial cells [20] and nasopharyngeal epithelial cells [21]. We found that METCAM is not expressed in most normal prostate epithelial cells [22] and breast epithelial cells [23].

Our initial work focused on mouse melanoma and the cloning of mouse METCAM cDNA [24]. We found that it may or may not promote the tumorigenesis of K1735-3 and -10 cell lines, but that it can promote the metastasis of these two cell lines [25]. However, it acts as a tumor suppressor and a metastasis suppressor for another mouse melanoma cell line, K1735-9 [26]. Thus, we first noticed that METCAM may play a dual role in the progression of cancers [18]. On the one hand, METCAM promotes the tumorigenesis and metastasis of breast cancer cells [23,27,28] and prostate cancer cells [29,30,31]. On the other hand, it suppresses the tumorigenesis and the progression of ovarian cancer cells [32]. It also plays a dual role in different types of NPC, for example, as a tumor suppressor for NPC type1 [33], and as a tumor promoter for NPC type III [34]. From our experience investigating its role in the progression of five different human cancers, we came to the conclusion that METCAM appears to play a dual role only in different cancer cell lines of the same cancer type or in different cell lines from different cancer types [35].

We pioneered research into METCAM’s role in the progression of prostate cancer. We cloned and sequenced the human METCAM cDNA from melanoma and from prostate cancer cell lines [12,24]. We used different portions of cDNA to overly express different portions of the protein in *E. coli* and used them to induce the production of polyclonal antibodies in chickens for the later immunological detection of its expression in various normal and cancer tissues [12]. From the immunohistochemical results, we found that most of the normal prostatic epithelium, and all of the benign prostatic hypertrophy (BPH), did not express huMETCAM, but that most prostatic intraepithelid neoplasia (PIN), high-grade prostatic carcinomas, and metastatic lesions expressed the protein [12,17,29]. The increased expression of METCAM was also similarly found in a transgenic model, TRAMP, during the malignant progression of mouse prostatic adenocarcinoma [36]. In summary, there is a positive correlation involving the over-expression of METCAM with the pathological grade of clinical prostatic carcinoma and with that of mouse adenocarcinoma in a transgenic mouse model, TRAMP. Taken together, we suggested that METCAM might be used as a possible new diagnostic marker for predicting the malignant potential of prostatic carcinoma [37]. The results of the following tests confirm this notion.

## 3. Using Various Immunological Methods to Validate METCAM as a Diagnostic Marker for Predicting the Malignant Propensity of Prostate Cancer

### 3.1. Use of Three Immunological Methods—Immunoblot (Western Blot), ELISA, and Traditional LFIA—To Test Whether METCAM May Serve as a Biomarker for the Malignant Potential of Prostate Cancer

First, we used the immunoblot (Western blot) method to test whether METCAM was present in the serum. We found that the METCAM was detectable in the serum samples and further that METCAM was significantly expressed at higher levels in prostate cancer tissues than in normal tissue and BPH [38]. Then, we used the more sensitive enzyme-linked immunosorbent assay (ELISA) method to quantitate the amount of METCAM and confirmed the results obtained by the Western blot method [38]. Furthermore, in an attempt to obtain more convincing data, we used the more simplified gold nanoparticle-based lateral flow immunoassay (traditional LFIA) [38,39,40] to increase the sensitivity of measurement, as shown in Figure 2 below. 

With this traditional LFIA, we not only confirmed the results of using the Western blot method and the ELISA but also obtained results revealing that the serum concentration of METCAM is statistically significantly higher in prostate cancer patients than that in normal individuals, thereby proving the notion that METCAM has high potential to serve as a new diagnostic marker for the detection of prostate cancer, as shown in Figure 3.

In addition, we showed that the serum concentration of METCAM is directly proportional to most of the serum PSA concentrations when PSA was <12 ng/mL [38]. To further improve the sensitivity, accuracy, and reproducibility of the results, we tried various alternative methods, as listed in the following subsections.

### 3.2. Using a Magnetic Bead-Enriched Array Display Immunoassay (MBIA) Method also Supports the above Notion

To decrease the possible interference from other substances in the serum samples so as to increase the sensitivity and specificity of the assay, we developed a magnetic bead-enriched array display immunoassay (MBIA) method by taking advantage of the extremely high affinity between biotinylated antibodies and streptavidin-coated magnetic beads to facilitate their interactions with the specific antigen and to amplify the signal by using cy5-secondary antibody for the test to further validate METCAM/MUC18’s role as a diagnostic marker for prostate cancer. In brief, the streptavidin-coated magnetic beads interacted with the biotinylated rabbit or chicken anti-METCAM Ab and then with the serum samples, which then interacted with rabbit or chicken anti-METCAM Ab, which, in turn, interacted with the cy5-goat anti-chicken or anti-rabbit Ab. Consequently, the cy5 was released by heating, spotted on a glass, and quantitated by a microarray confocal scanner [43], as shown in Figure 4.

Using this MBIA, we also confirmed the above results, as shown in Figure 5 below.

### 3.3. Using a Modified LFIA Method Not only Supports the above Notion but also Further Shows That METCAM Can Be Used to Predict Malignant Propensity at the Pre-Malignant (Pin) Stage

We further improved the traditional LFIA test and thus developed the modified LFIA test to obtain better data with the lowest standard deviation. For this, we incorporated biotin and streptavidin into the traditional LFIA. In brief, the test line on the nitrocellulose membrane was sprayed with streptavidin to capture the triple complexes of biotinylated rabbit or chicken anti-METCAM Ab/METCAM antigen/nano-gold conjugated chicken or rabbit anti-METCAM Ab. The control line on the nitrocellulose membrane was sprayed with goat anti-chicken or rabbit Ab to capture the excess nano-gold-conjugated anti-METCAM Ab [44,45], as shown in Figure 6 below.

### 3.4. Using Magnetic-Beads to Purify the METCAM from Serum Samples and, Subsequently, Subject Them to the Modified LFIA, which also Supports the above Notion

To decrease the effects of interfering substances in human serum, we improved the sensitivity and specificity of the detection method, and used the streptavidin-coated magnetic beads and biotinylated anti-METCAM antibodies to purify the METCAM from human serum samples [46]. The streptavidin-coated magnetic beads were allowed to interact with the biotinylated anti-METCAM antibody; subsequently, the complex was incubated with the serum samples. The attached METCAM antigen was then eluted and applied to the assembly of the modified LFIA. The results are shown in Figure 7 below.

We also added a silver enhancing reagent to the LFIA to enhance the detection signal; consequently, we found that the signal had increased at least one- to two-fold [46].

### 3.5. Direct Application of Triple Complexes to the Modified LFIA

Instead of using magnetic-bead-purified serum samples for analysis in the modified LFIA, we directly applied the triple complexes of biotinylated anti-METCAM Ab-antigen-nano-gold-conjugated second anti-METCAM Ab to the modified LFIA assembly [41,44,45]. By using this simplified method, we showed that METCAM has high potential with respect to the prediction of the malignant potential of prostate cancer at the early premalignant PIN stage, as shown in Figure 8.

The following Table 1 summarizes the pros and cons of the various immunological methods for the determination of serum huMETCAM concentrations.

Taken together, we have used various immunological methods to consistently prove and confirm the notion that METCAM may serve as a biomarker for predicting the malignant potential of prostate cancer. Furthermore, by improving the sensitivity and specificity of these methods, we suggest that METCAM may be used as a biomarker for predicting the malignant potential of prostate cancer at the pre-malignant (PIN) stage. In contrast, we have firmly proven that the use of PSA has failed with respect to predicting the malignant potential of prostate cancer, as shown in the following Figure 9.

In summary, our preclinical research has shown that METCAM likely offers superior potential compared to the current PSA test with respect to acting as a diagnostic biomarker for the prediction of the malignant potential of prostate cancer at the early stage; however, this notion needs to be validated after clinical trials.

## 4. METCAM Plays a Positive Role in the Malignant Progression of Most Prostate Carcinoma Cells

We were the first group to demonstrate that the increased expression of METCAM in a human prostate cancer cell line, LNCaP, enhances its EMT (epithelial-to-mesenchymal transition) activity, as shown in analyses of augmented motility and invasiveness. Furthermore, the increased expression of the protein enhances the tumorigenesis in vivo and starts its spread to multiple organs after injecting the cells in the orthotopic route (prostate gland) in a nude male mouse model. Taken together, we suggested that METCAM is a genuine metastasis gene, which is capable of initiating and promoting the malignant progression of human prostate cancer cells [29,30,31]. The positive role of METCAM in the progression of prostate cancer is also demonstrated in cell lines other than LNCaP, such as DU145 [37]. Thus, METCAM may be used to differentiate aggressive prostate cancer from the corresponding indolent cancer.

## 5. METCAM as a Therapeutic Target for Clinical Treatment of Patients with Malignant Prostate Cancer

Since METCAM is an authentic driver in the progression of human prostate cancer cells, as shown above, it is highly likely that the anti-METCAM antibody may be used to arrest the spreading of cancer cells [42,47,48].

In addition, we have shown that three METCAM-specific shRNAs, which are expressed in a lentivirus vector, could decrease the tumorigenesis of another human prostate cancer cell line, DU145, which endogenously expresses a high level of METCAM, in Balb/C athymic nude male mice [37], as shown in Figure 10 below.

Thus, we suggest that it is highly likely that METCAM-specific shRNAs in lentivirus vectors will be used for therapeutic treatment to arrest the malignant progression of clinical prostatic carcinoma.

Moreover, we also showed that soluble METCAM could block the angiogenesis of LNCaP tumors in a pre-clinical, athymic, nude mouse model [49]; thus, METCAM-derived oligo-peptides or a soluble form may be used to arrest the progression of prostate cancer [42,49].

From our mechanistic studies, it has been determined that METCAM may modulate these processes by augmenting proliferation, increasing the influence of the AKT-signaling pathway, boosting aerobic glycolysis, and increasing the angiogenesis of prostate cancer cells; however, it has no effect on apoptosis. In the future, these downstream effectors may be used as targets for designing therapeutic agents to block the progression of this cancer [50].

## 6. Conclusions and Perspectives

We have discovered and presented evidence that METCAM, a cell adhesion molecule in the Ig-like superfamily, has high potential in terms of its use as a new biomarker for the prediction of the malignant propensity of prostate cancer. Since METCAM has also been shown to be able to initiate the spreading of prostate cancer cell lines to multiple organs, we also suggest that it may be used as a therapeutic target for designing therapeutic means to arrest the malignant progression of prostate cancer or the clinical treatment of patients with malignant prostate cancer. Four possible strategies may be used for clinical treatment: (a) humanized anti-METCAM antibodies, (b) METCAM-specific shRNAs in a lentivirus vector, (c) METCAM-derived oligo-peptides or in a soluble form, and (d) the blocking of the cognate heterophilic ligand(s) and the expression of the METCAM-downstream effectors.

Autophagy plays a pivotal role in the regulation of apoptosis and the disease progression of prostate cancer, and it involves several molecular pathways, including PI3K/AKT/mTOR; however, the role of autophagy in this cancer is complex [51,52,53]. This is because autophagy may play either a detrimental or a protective role in prostate cancer, depending on the cellular context, in that it may serve as a tumor suppressor during the early stages of tumor cell development in order to constrain cancer initiation and later as a pro-tumorigenic process [51,52,53], in which autophagy may provide necessary nutrients and manage ROS to support tumor growth. Further evidence also shows that therapeutic resistance to cytotoxic chemotherapy, molecularly targeted agents, and radiotherapy may also be supported by autophagy. However, stronger evidence is still needed to better understand the role of autophagy in the progression of prostatic cancer, and the application of these findings in clinical practice may be forthcoming in the future [51,52,53]. How METCAM uses autophagy to manifest its role in the progression of the prostate cancer is not known. This aspect constitutes an interesting research aspect to be explored in future studies.

## Figures and Tables

**Figure 1 biomedicines-11-00205-f001:**
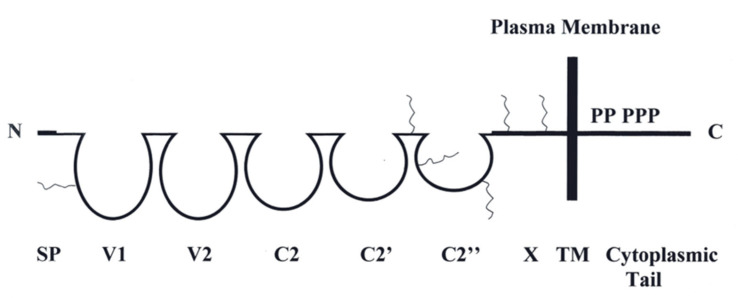
The protein structure of huMETCAM. In the extracellular part of the protein: SP stands for the signal peptide sequence at the N-terminus; five domains, V1, V2, C2, C2′, and C2″, stand for five immunoglobulin-like domains (which are held by a disulfide bond); and one domain, X, stands for another domain (without a disulfide bond). TM stands for the transmembrane domain. P stands for the five possible phosphorylation sequences (one for PKA, three for PKC, and one for CK2) in the cytoplasmic tail of the protein. The wiggly lines indicate the six conserved N-glycosylation sites located in the V1 domain, the region between the domains of C2′ and C2″, the domain of C2″, and the X domain.

**Figure 2 biomedicines-11-00205-f002:**
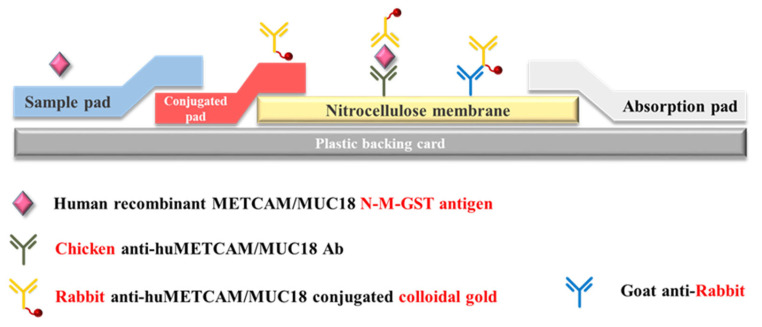
The procedure of the traditional LFIA [41].

**Figure 3 biomedicines-11-00205-f003:**
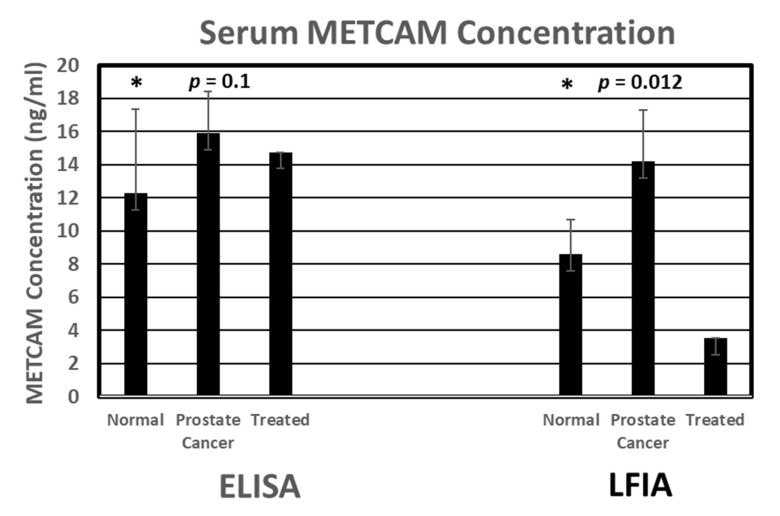
The serum METCAM concentrations determined by gold nano-particle-based LFIA in comparison with ELISA [42]. Consequently, it was revealed that METCAM was highly likely to serve a new biomarker for the diagnosis of the malignant potential of prostatic carcinomas.“*” is the reference data for statistical analysis.

**Figure 4 biomedicines-11-00205-f004:**
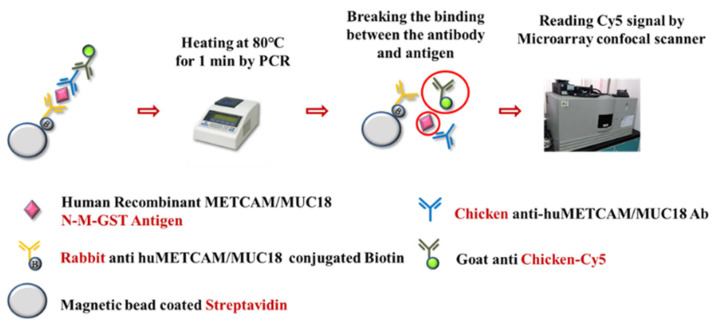
The procedure of MBIA [43]. The large circle indicates the released Cy5-conjugated goat anti-chicken and the small circle indicates the released antigen after heating.

**Figure 5 biomedicines-11-00205-f005:**
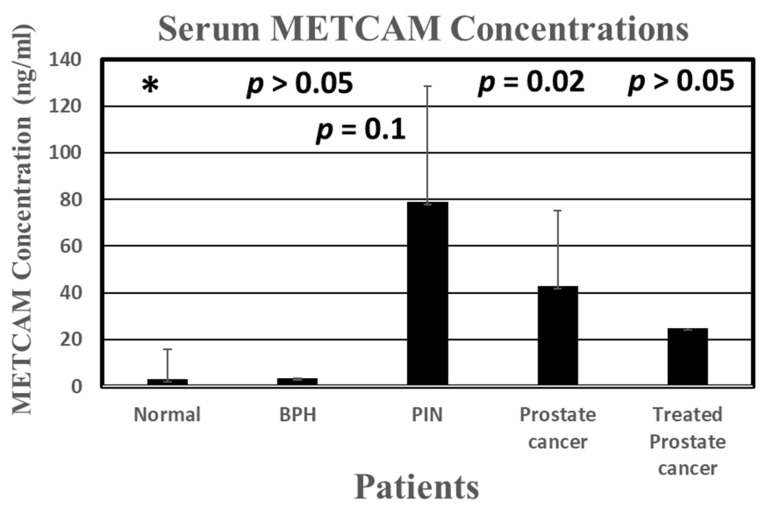
Serum METCAM concentrations in normal individuals and patients with BPH (benign prostatic hypertrophy), PIN (prostatic intraepithelial neoplasia), prostate cancer, and various treated patients as determined by the MBIA method [43].“*” is the reference data for statistical analysis.

**Figure 6 biomedicines-11-00205-f006:**
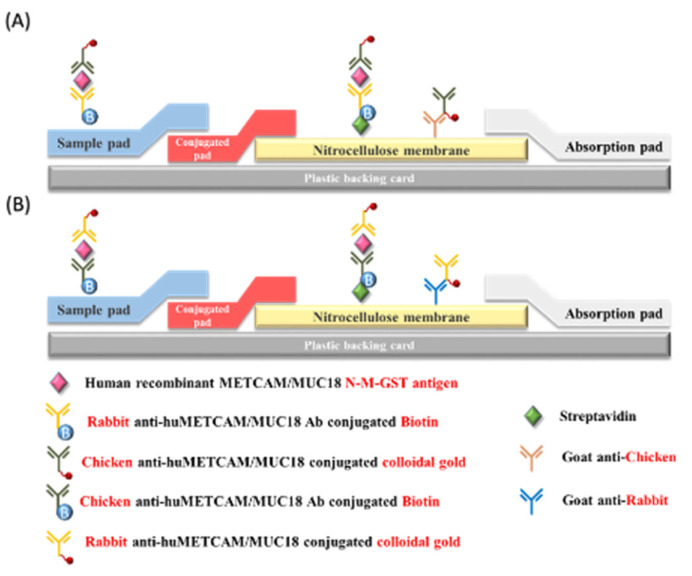
The procedure of the modified LFIA [44,45]. (**A**) indicates the antibody combination of biotinylated rabbit antibody and the nano-gold-conjugated chicken antibody. (**B**) indicates the antibody combination of biotinylated chicken antibody and the nano-gold-conjugated rabbit antibody.

**Figure 7 biomedicines-11-00205-f007:**
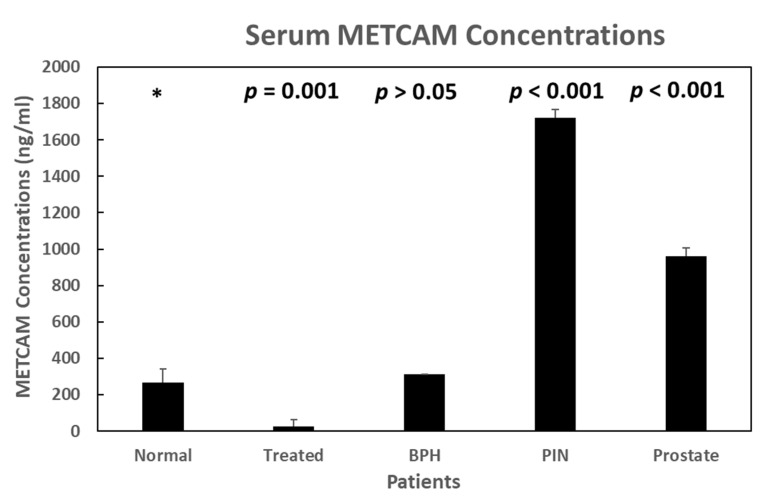
Serum METCAM concentrations in normal individuals and patients of BPH, PIN, prostate cancer, and treated were determined after magnetic-bead purification and then followed by the modified LFIA method [46]. “*” is the reference data for statistical analysis.

**Figure 8 biomedicines-11-00205-f008:**
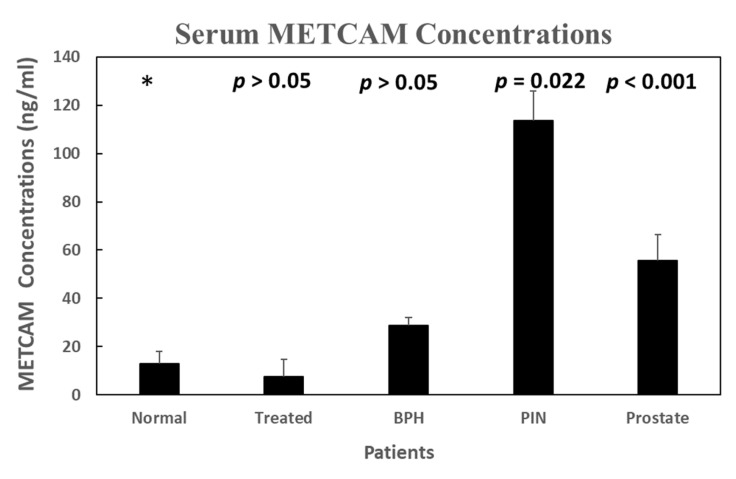
Serum METCAM concentrations in normal individuals and patients with BPH, PIN, prostate cancer at different Gleason scores (GS), and treated as determined by the direct application of the triple complexes to the modified LFIA [41,44,45]. ”*” is the reference data for statistical analysis.

**Figure 9 biomedicines-11-00205-f009:**
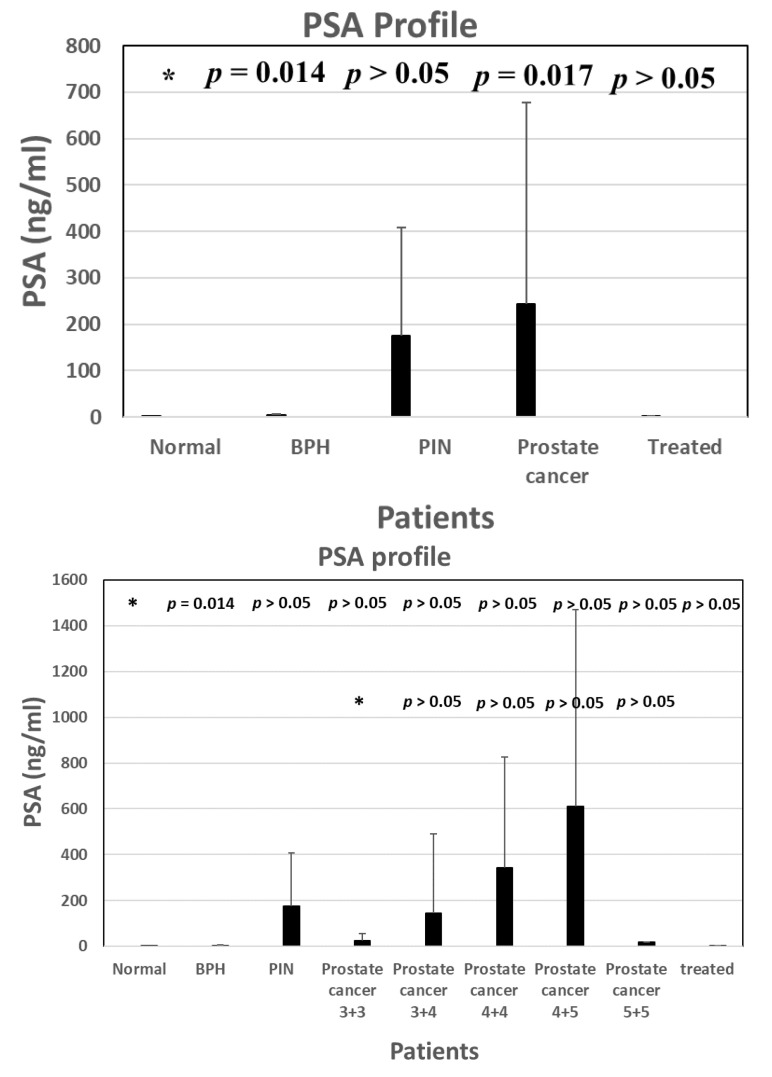
PSA levels in normal individuals and patients of BPH, PIN, and prostate cancer and treated prostate cancer patients. The numbers (3 + 3 etc.,) under each category of prostate cancer are Gleason scores. ”*” is the reference data for statistical analysis.

**Figure 10 biomedicines-11-00205-f010:**
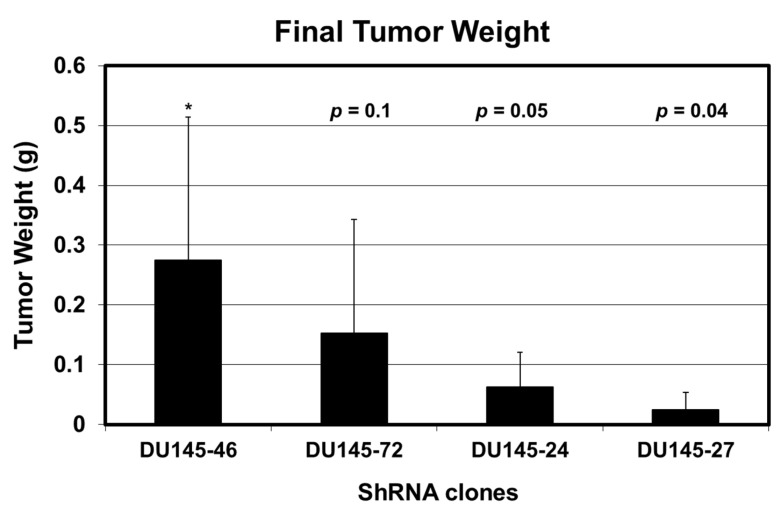
Tumorigenicity of METCAM-specific shRNA-transfected DU145 clones. Three METCAM-specific shRNA #72, 24, and 27 in lentivirus vector blocked the tumorigenesis of DU145 cells in a mouse model. ShRNA 46 is a non-METCAM shRNA control [37]. “*” is the reference data for statistical analysis.

**Table 1 biomedicines-11-00205-t001:** Comparison of various immunological detection methods.

Detection Method	Advantages	Disadvantages
Western blot analysis	Molecular sizes can be determined	Labor-intensive and results may vary depending on workers. Electrophoresis and electric-blotting equipment are needed. Not sensitive enough.
ELISA	More sensitive than WB	Labor-intensive, results vary depending on workers, expensive reader and other equipment for quantitation is needed.
Traditional LFIA	More sensitive, accurate, and reproducible than WB and ELISA	Not labor-intensive, results were not easily reproducible.
MBIA	Many samples can be quantitated easily on a single glass slide	Results vary among different workers; results were not easily reproducible
Modified LFIA	More sensitive and accurate than traditional LFIA; results are easily reproducible	Not many
Magnetic bead-purified serum + modified LFIA	Purification step can be used to remove interfering contaminants in the serum samples	Purification step may not be easily reproducible
Triple complexes + modified LFIA	More sensitive and accurate than traditional LFIA, results are easily reproducible, simple, inexpensive	Not many

## Data Availability

All the data used in these studies are available upon request to the corresponding author (GJW).

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
