# Peer review of "METCAM Is a Potential Biomarker for Predicting the Malignant Propensity of and as a Therapeutic Target for Prostate Cancer"

_biomedicines, 2023, doi:10.3390/biomedicines11010205_

Round 1
Reviewer 1 Report
Authors used various immunological methods to consistently prove and confirm the notion that METCAM may serve as a biomarker for predicting the malignant potential of prostate cancer. They also suggested possible strategies for clinical treatment using this target. The topic is wontly of interest and the manuscript has various strengths. However, it needs a syntax check, and it is lacking in several points that would add value to the entire manuscript: a major revision is required.
- I suggest excluding this sentence from the abstract “The PSA, a serine pro- tease, is specifically expressed in prostate gland, thus an organ specific antigen, and the elevated serum PSA level has been used in the past to predict the malignant potential of prostate cancer.” This is because it is needed to be focused on the topic.
- Please reformulate this sentence in the introduction “In fact, most prostate cancer patients die with the cancer, not of, nor from the cancer, thus, no need for treatment.” However, a reference it is needed here.
- Please vide a full definition of METCAM before using it in the introduction.
- Please provide a better image quality for Figure 1 and Figure 10.
- When you said: “we have shown that METCAM is superior to the current PSA test to be used as a diagnostic biomarker for predicting the malignant potential of prostate cancer.”, this may be too extremes and need to be re formulated also considering the absence of clinical application of your biomarker.
- In the conclusion, please eliminate the four possible strategies. Moreover, they are only
- Even if you focused on diagnostic markers for predicting malignant prostatic carcinoma cancer, I strongly believe it is worthy of interest considering markers to predict radicality after RARP and maybe recurrence. Many biomarkers shave been proposed for the scope and one of the latest is urinary levels of 8-OHdG and 8-iso-PGF2α. Authors showed how this helps predict radicality (and perhaps local recurrence). Please include this paper and discuss is in your paper (doi: 10.3390/jcm11206102; PMCID: PMC9605140; PMID: 36294423).
- When talking about biomarkers, tumor microenvironment you should be consider. It is well known that autophagy plays a crucial role in cancer development and in response to treatments. Recent findings have been conducted on molecular and biological pathways of autophagy. Please provide a brief description and include this new paper (doi: 10.3390/ijms23073826; PMCID: PMC8999129; PMID: 35409187).
- A syntax check is required.
Reviewer 2 Report
The manuscript by Wu and Wu described that METCAM, a cell adhesion molecule in the Ig-like superfamily, has a high potential to be used as biomarker and therapeutic target of prostate cancer.
The manuscript is interesting, but some major points need to be addressed:
1) Please, the manuscript needs to be carefully revised for English language
2) The title must be clear, concise and informative. It could be better to change the title in: METCAM is a potential biomarker and as a therapeutic target for prostate cancer
3) Please, divide the manuscript in the following sections: introduction, materials and methods, results, discussion, conclusions
4) You need to completely re-write the introduction section, including a detailed overview of the landscape of tumor markers in prostate cancer. See, cite and discuss the following article:
Ferro M, De Cobelli O, Lucarelli G, Porreca A, Busetto GM, Cantiello F, Damiano R, Autorino R, Musi G, Vartolomei MD, Muto M, Terracciano D. Beyond PSA: The Role of Prostate Health Index (phi). Int J Mol Sci. 2020 Feb 11;21(4):1184. doi: 10.3390/ijms21041184. PMID: 32053990; PMCID: PMC7072791.
5) Please, detail the criteria to identify a clinically significant prostate cancer
6) Figure of METCAM is a bad quality figure
7) Put p values in the text and not in the figures
8) Perform a ROC curve analysis to compare the diagnostic performance of METCAM compared to PSA and possibly to PHI in the ability to discriminate between BPH and prostate cancer and between prostate cancer with GS<7 versus GS> or = 7
Round 2
Reviewer 1 Report
The manuscript has been deeply improved. I believe it is worthy of publication in its current form.
Author Response
Please see the attchment.

Reviewer 2 Report
The authors address all the key points and the manuscript can now be accepted in its present form
Author Response
lease see the attachment.
